# Laparoflow-SSL: Image Analysis From a Tiny Dataset Through Self-Supervised Transformers Leveraging Unlabeled Surgical Video

**Karel Moens**[1,2]                                        karel.moens@esat.kuleuven.be

**Jonas De Vylder**[2]                                       jonas.devylder@barco.com

**Matthew B. Blaschko**[1]                              matthew.blaschko@esat.kuleuven.be

**Tinne Tuytelaars**[1]                                     tinne.tuytelaars@esat.kuleuven.be

[1] *Dept. of Electrical Engineering, PSI, KU Leuven, Belgium*

[2] *Barco NV, Belgium*

## Abstract

During minimally invasive surgery, surgeons monitor their actions and the relevant tissue through a camera. This provides an ideal environment for artificial intelligence (AI) assisted surgery. For the development of such AI components, the need for expert annotations remains a key bottleneck. In this paper, we study the application of self-supervised learning (SSL) on surgical data. In a self-supervised setting, a representation backbone is trained on information that is inherently present in the data. There is no need for annotations, leaving the backbone free to train on all recordings, not just labeled ones. We leveraged optical flow for weighting pairs in a view-contrastive self-supervised learning loss. Constructed as an Info Noise-Contrastive Estimation (InfoNCE) loss, it contrasted the pixel representations of two differently, photometrically and geometrically transformed views. The importance of each contrasted pixel pair is determined by computing the difference between the optical flows of the respective pixels. In this way, the optical flow guided the representations of pixels that move together to similar vectors. We tested the usefulness of the representation vectors by training simple networks for semantic segmentation or robotic instrument key point detection. These networks showed competitive performance, even when using 93% fewer annotated samples than other works. For semantic segmentation, we used as little as 99.73% fewer samples for training, originating from the m2caiSeg dataset, and remained competitive even when testing on the unseen cholecSeg8k dataset.

**Keywords:** Data Efficiency, Laparoscopic Segmentation, Instrument Pose, Self-Supervision.

## 1. Introduction

Minimally invasive surgery reduces incisions on a patient, resulting in a shorter recovery time. As an added benefit, it also requires the use of a camera and therefore digitizes the visual inputs to the surgeon. Increasingly relevant in this age, is that the digitized videos allow computers to learn from the surgery. However, deep learning is data hungry, which is problematic because annotation of medical structures is very expensive. With video data always available during minimally invasive surgery, we propose, therefore, to utilize all video data, even unlabeled, for pretraining of deep learning models through two contrastive self-supervised learning signals. First, we leverage photometric and geometric invariance for all video frames. Second, we extract motion information to infer object boundaries and coherence. In what follows, we show that by integrating a representation backbone that was

pretrained in such a way, it is possible to construct competitive downstream task specific models with a fraction of the annotated data. Our results indicate that this holds even when these models are tested on images from different medical institutes.

Previous works directly combined optical flow in surgical video with semantic masks (Zhao et al., 2020; Sestini et al., 2023; Alapatt et al., 2021). This strongly couples the training phases and is more restrictive to the task of segmentation of the chosen classes. Others leave motion unused (Ross et al., 2018). Mahendran et al. (2019) work on natural videos (Van Gansbeke et al., 2021; Du et al., 2022) and regress their representations to a kernel applied to the optical flow, strongly enforcing distances and leaving little room for other learning signals. A typical use of optical flow is for tracking. Takahashi et al. (2023),Sharma et al. (2022), Xiong et al. (2021) and Wang and Gupta (2015) construct positive pairs from pixels tracked through time for self-supervision, as opposed to finding pairs within a video frame. Many works have explored weak- or self-supervision for medical imaging (Chaitanya et al., 2020; Hu et al., 2021; der Sluijs et al., 2023; Wang et al., 2022). To our knowledge, the use of optical flow during self-supervision to contrast parts of images has remained unstudied. Our contributions are the following:

- We provide further exploration of self-supervised learning on laparoscopic video.

- We propose novel, dense pixel representation contrastive loss functions that take advantage of the availability of surgical video. The optical flow derived from video is transformed into weights that enable mining of positive and negative pixel pairs within video frames. This mining is theoretically compatible with many contemporary methods.

- We describe network architectures for downstream tasks that integrate large representation backbones, irrespective of the backbone architecture, and, as a result, only require a tiny annotated dataset for fine-tuning while demonstrating resilience to overfitting.

The Method section provides a top-down explanation of our proposed loss function and the chosen network architectures. The Experiments section details the datasets, hyperparameters and measured performance.

## 2. Method

### 2.1. Weighted sum of all pixel triplets.

For our loss function, we adapt the InfoNCE formula (Oord et al., 2018) for representations of pixels instead of entire images, by assigning each pixel in an image a positive and negative weight with respect to every other pixel as anchor. These weights quantify how well the pair of pixels fits as a positive pair or a negative pair. The network is rewarded when representations of pairs from the positive set $\mathcal{P}_{i,a}$ for anchor pixel $a$ in image $\mathbf{x}_i$ are similar and when the representations of pairs from the negative set $\mathcal{N}_{i,a}$ are dissimilar. As shown in Equation (1), the entire loss term is the mean of InfoNCE over all images $\mathbf{x}_i$ in the dataset $\mathcal{D}$, applied to all pixel triplets (anchor, positive, negative) multiplied by both the positive and the negative weights. As proposed by Pinheiro et al. (2020), we give negative similarities

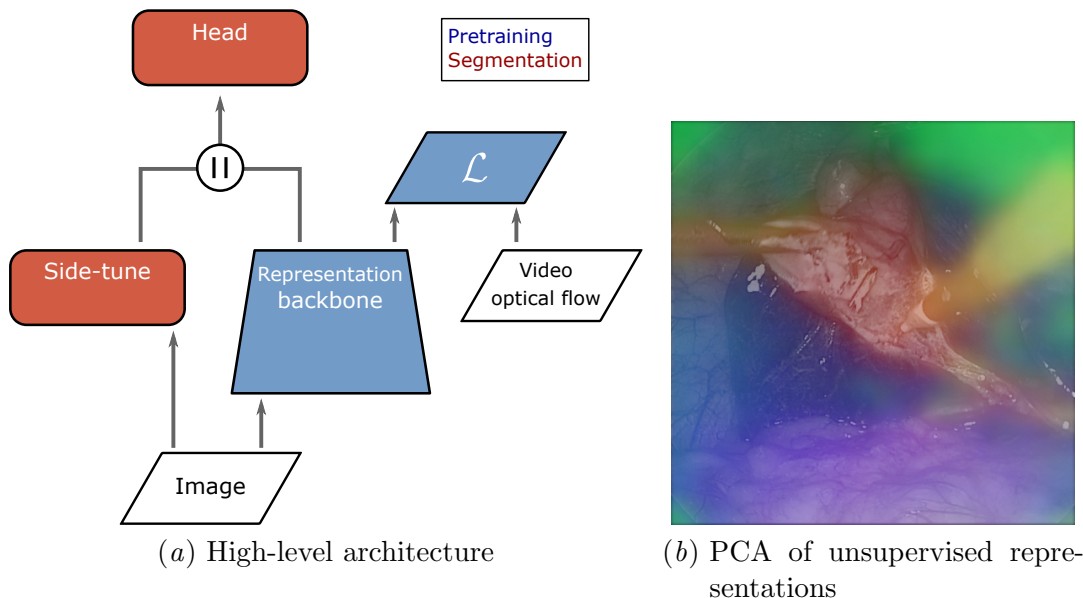

(*a*) High-level architecture

(*b*) PCA of unsupervised representations

Figure 1: (*a*) During pretraining, the pixel representation backbone is contrastively trained on unlabeled video frames, leveraging invariance to transformations and guided by optical flow vector distances. When fine-tuning, the backbone parameters are frozen and only a shallow side-tune network and head are trained on a tiny dataset of annotated images. (*b*) After pretraining, we can reduce the dimensionality using principal component analysis (PCA) to the GRB color channels. No ground truth semantic masks or explicit clustering was involved. The gall-bladder pixel representations align with the second component, giving them a red hue. Surgical instruments color yellow and orange, fat tissue purple and the liver appears to be more blue. The primary component (green) makes a distinction between black background pixels and others. Note that the pixel representations were scaled up from $64 \times 64$ resolution.

in the denominator a factor of 10 more weight to avoid the failure mode of outputting the same representation vector for all input pixels. The next paragraphs provide more details on the similarity function $s_{i,a,p}^{(u,v)}$ and the weights.

$$\mathcal{L} = - \mathop{\mathbb{E}}_{\mathbf{x}_i \in \mathcal{D}} \left[ \sum_{a \in \mathbf{x}_i} \sum_{p \in \mathcal{P}_{i,a}} w_p \sum_{n \in \mathcal{N}_{i,a}} w_n \log \left( \frac{s_{i,a,p}^{(u,v)}}{s_{i,a,p}^{(u,v)} + 10 \cdot s_{i,a,n}^{(u,v)}} \right) \right] \tag{1}$$

### 2.2. Video frame view generation.

A common technique in self-supervised representation learning is view contrasting. When applied, two views of a dataset sample are generated so that both views retain their task specific semantics. This is similar to the use of augmentations during training or at test-time. The representations of the two views are contrasted and a loss is calculated to penalize

differences, since the task semantics are the same. Inspired by the equivariance described by Cho et al. (2021), we use photometric and geometric transformations to generate the views. Photometric transformations stimulate invariance to endoscope properties and lighting. Geometric transformations stimulate invariance to viewing angle and instrument pose. Figure 2 (appendix) gives a visualization and more information on the composition of these transformations. Equations (2)-(4) describe the pixel representations of the different views.

$$\mathbf{z}_{i,k}^{(1)} = f_\theta\Big(G_i^{(1)}\big(P_i^{(1)}(\mathbf{x}_i)\big)\Big)\Big[k\Big], \tag{2}$$

$$\mathbf{z}_{i,l}^{(2)} = f_\theta\Big(G_i^{(2)}\big(G_i^{(1)}\big(P_i^{(2)}(\mathbf{x}_i)\big)\big)\Big)\Big[l\Big], \tag{3}$$

$$\mathbf{z}_{i,l}^{(3)} = G_i^{(2)}\Big(\mathbf{z}_i^{(1)}\Big)\Big[l\Big], \tag{4}$$

with $\mathbf{x}_i$ an input image, $f_\theta$ the representation backbone, $P$ and $G$ random photometric and geometric transformations, and $k$ and $l$ indices of pixels in the image. In the implementation of these composite functions, we enforce all $||\mathbf{z}||_2 = 1$ to reduce the cosine similarity to a dot product. Notice that the same composite geometric transformation is applied to $\mathbf{z}_i^{(2)}$ and $\mathbf{z}_i^{(3)}$ but that the view that passes through the representation backbone $f_\theta$ is different. The representations are compared through the similarity function from Equation (5) with $\tau_0 = 0.07$, as is common for visual InfoNCE (Pinheiro et al., 2020; He et al., 2020).

$$s_{i,k,l}^{(u,v)} = \exp\left(\frac{\langle\mathbf{z}_{i,k}^{(u)}, \mathbf{z}_{i,l}^{(v)}\rangle}{\tau_0}\right). \tag{5}$$

## 2.3. Optical flow guided pixel representation learning.

When perceiving motion, things that move similarly are likely to be correlated, perhaps even part of the same object. This is the reasoning behind the optical flow guided pixel representation learning. Given the optical flow for every pixel in an image, we calculate the Euclidean distances between these vectors as an approximation of belonging to the same object. When the distance between flow vectors is small, it is likely that they are part of the same object. Conversely, when the distance is large, the pair of pixels should be treated as negatives for each other. This hypothesis is visualized in Figures 9 and 10 in the appendix. In our implementation, we use half of the pixels in every frame as positives for a given anchor, and half as negatives, with the median distance as a threshold, Equations (6) and (7). Of course, within these halves, some pairs of pixels will be more fitting as positives or negatives than others. We therefore calculate the weight of each pair based on the optical flow distance. Simply relying on Euclidean distance for the calculation of positive and negative weights can be problematic. Because some objects move more than others, like instruments in a surgical setting, the distances to the pixels that comprise these objects will be many times larger than between pixels of objects with more subtle motion. As a result, pair weights based on Euclidean distance of flows can collapse into merely contrasting objects with potentially strong motion against everything else. We propose to mitigate this by calculating the relative distance between motion flows. As the distance is then relative to either of the norms of the involved vectors, this distance is no longer symmetric. As described in Equation (8), we normalize the Euclidean distances with the norm of the flow

vector $\mathbf{f}_i$ of the anchor pixel in each anchor-positive-negative triplet. A consequence of this decision is that relative distances to anchor pixels that barely move can become very large. That is why a threshold is introduced for ignoring anchor pixels with too little motion. These quasi-stationary pixels still appear as positives and negatives with respect to other anchors, therefore their representations will still be shaped by the optical flow guided loss function. In our experiments, we set this threshold to $\frac{1}{3}\frac{\text{pixel}}{s}$. Figure 8 in the appendix shows some examples of pixels in the positive and negative set. To bound the weights between $[0,1]$, we apply the exponential function to the negative of the relative distance. Finally, we introduce temperature parameters $\tau_P$ and $\tau_N$ for the positive and negative weight calculations and set these, in line with the median heuristic (Gretton et al., 2012), to the 25% and 75% quantiles of the relative distances in the unlabeled pretraining set, respectively. As the only certain positive pair, we assign extra weight to the corresponding pixel in the other view. For all reported results of Laparoflow-SSL, this cross-view exact positive match weight is set to 4. Our final loss function is then given by Equation (1) with similarity function $s^{(2,3)}$ (Equations (5), (3) and (4)), and $w_p$ and $w_n$ as in Equation (9). In the following equations, $i$ indicates an image and $k$, $l$, $a$, $p$ and $n$ refer to pixels in that image.

$$\mathcal{P}_{i,a} = \left\{ p \mid \Delta f_{i,a,p} \leq Q_{\Delta f_{i,a}}(50\%) \right\}, \tag{6}$$

$$\mathcal{N}_{i,a} = \left\{ n \mid Q_{\Delta f_{i,a}}(50\%) < \Delta f_{i,a,n} \right\}, \tag{7}$$

$$\Delta f_{i,k,l} = \frac{\left\| G^{(1)}(\mathbf{f}_i)[k] - G^{(1)}(\mathbf{f}_i)[l] \right\|_2}{\left\| G^{(1)}(\mathbf{f}_i)[k] \right\|_2}, \tag{8}$$

$$w_p = \exp\left(\frac{-\Delta f_{i,a,p}}{\tau_P}\right), \quad w_n = 1 - \exp\left(\frac{-\Delta f_{i,a,n}}{\tau_N}\right). \tag{9}$$

## 2.4. Network architecture.

The proposed self-supervised learning approach is independent of the network architecture. For the construction of pixel representations, the architecture of any semantic segmentation model can function as a starting point. For our experiments, we chose the smallest SegFormer architecture (Xie et al., 2021). SegFormer is a hybrid, lightweight Transformer architecture that was designed for real-time semantic segmentation. After removing the softmax over the classes in the final layer, the dimensions of the outputs of that layer can be modified to the desired representation dimensions. We use representation of size 256.

In our experiments, we freeze the parameters of the representation backbone. When training on a downstream semantic segmentation task, a 2 layer concatenating side-tune network (Zhang et al., 2020) is placed parallel to the chosen backbone architecture (Figure 1). For key point detection, a quick exploration indicated that this task benefits more from narrow heads rather than shallow ones. It is outside the scope of our study to implement components included by Du et al. (2018), such as edge probability maps and bipartite matching. These components are complementary, however, and we expect them to bring further improvements. For additional details or reasons behind the downstream architectures, we refer the reader to Appendix D.

## 3. Experiments

### 3.1. Datasets.

#### 3.1.1. Self-supervised representation backbone pretraining.

For the training of the representation backbone to extract meaningful pixel representations, we compose a dataset of laparoscopic and robotic surgery videos. The laparoscopic videos originate from the M2CAI 2016 challenge (Twinanda et al., 2016; Stauder et al., 2016). The RARP-45 dataset (Van Amsterdam et al., 2022) serves as the source for the robotic surgery videos. Since the latter dataset is smaller, we repeat frames extracted from RARP-45 videos 5 times to have more or less equal occurrences of laparoscopic and robotic instruments in our pretraining dataset.

#### 3.1.2. Semantic segmentation.

Since every medical institute has some differences in optical and hardware stack, we chose to recreate this setting by using one dataset for pretraining and downstream fine-tuning, and another for testing. To train the segmentation heads and models, we used the m2caiSeg dataset (Maqbool et al., 2020), based on M2CAI. Cholec80 (Twinanda et al., 2016) was chosen as the test domain for semantic segmentation, a popular laparoscopic surgery dataset that includes 80 video recordings. CholecSeg8k (Hong et al., 2020) is a dataset that builds on Cholec80 by adding semantic mask annotations of 13 classes to 8080 video frames of 101 clips extracted from 17 of those videos. Among the 15 videos in M2CAI, only some are shared with Cholec80. More importantly, M2CAI does not contain data from all contributing institutes of Cholec80. m2caiSeg contains 245 annotated images in its training set and 62 in its test set, all extracted from 2 videos. These 2 videos are not included in the pretraining nor are any of their frames annotated in CholecSeg8k.

The sets of annotated classes overlap between CholecSeg8k and m2caiSeg, allowing transfer of models trained on this intersection. Important for this study, the intersection includes tissue classes to verify that the self-supervision based on optical flow not only benefits the more noticeably moving instrument classes. We aggregate all instruments into one class and chose to include the gall-bladder, as primary manipulated object of the cholecystectomy, and fat, a tissue type that can appear in several shapes and places in video frames. An additional consequence of the transfer between datasets is that the annotators for the training and test set are most likely different.

#### 3.1.3. Pose estimation of robotic instruments.

As a second downstream task, different from semantic segmentation, we chose pose estimation of robotic instruments. During our experiments, we evaluate on a refined version of the EndoVis15 dataset (Du et al., 2018; EndoVis, 2015). From its 4 videos for training and 6 videos for testing on robotic instruments, we get 940 and 904 annotated frames respectively.

Table 1: Segmentation results for classes fat tissue (Fat), instruments (Ins), gall-bladder (Gb) and background (Bkg). The bottom rows are adopted from Silva et al. (2022), retaining significance and bold typesetting. The asterisk indicates that these are included to give a sense of scale, since the models were completely trained on all classes and data from CholecSeg8k. LSSL is an abbreviation for Laparoflow-SSL. All 'SSL' entries, including 'No SSL', have the same architecture. 'No SSL' was entirely randomly initialized and fine-tuned all parameters.

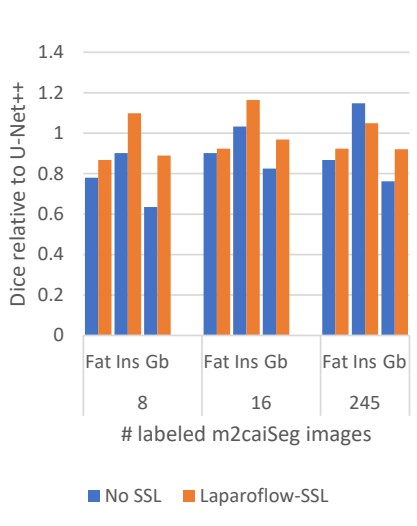

| #labeled *m2caiSeg* images for fine-tuning | | Test Dice on *CholecSeg8k* | | | |
|---|---|---|---|---|---|
| | | Fat | Ins | Gb | Bkg |
| 8 | No SSL | .71 | .55 | .40 | .81 |
| | Laparoflow-SSL | **.79** | **.67** | **.56** | **.83** |
| 16 | No SSL | .82 | .63 | .52 | .84 |
| | Laparoflow-SSL | .84 | .71 | .61 | .88 |
| | Stronger augment LSSL | .83 | .73 | .60 | .88 |
| | LSSL Cholec80 pretrain | **.86** | **.76** | **.62** | **.89** |
| | View-contrast, no flow | .82 | .59 | .48 | .83 |
| 245 | No SSL | .79 | **.70** | .48 | .82 |
| | Laparoflow-SSL | **.84** | .64 | **.58** | **.86** |
| 6k* | U-Net++ (Zhou et al., 2018) | **.91** | .61 | **.63** | |
| | (Hatamizadeh et al., 2022) | .88 | **.71** | .42 | |

## 3.2. Results.

Tables 1 and 2 list semantic segmentation and key point detection results respectively. Segmentation models built on Laparoflow-SSL are competitive, despite using annotated datasets of less than 0.27% the size, from different institutes than the test set. As proven by the entry with Cholec80 pretraining, even better performance can be achieved when one can also collect unlabeled videos from other domains than the annotation source. Stronger augmentations during pretraining can partially imitate this, though this is more effective when there is no domain transfer between train and test set, as can be seen when comparing the entries from Table 1 and Table 4 in the appendix.

For fat tissue, a gap still remains between the tiny annotated dataset and the full dataset setting. As patches of fat can be scattered and relatively small, the image resolution of $256 \times 256$ for these experiments leading to $64 \times 64$ pixel representations might be too rough. Another explanation is that this tissue type is not as coherent or rigid as instruments and organs. This is supported by the performance of pretrained models that did not use any optical flow. The larger drop in instrument or gall-bladder Dice compared to fat tissue, indicates that the latter benefits less from the learning signal derived from optical flow.

Table 2: The performance of the robotic instrument key point detection networks, as measured by Du et al. (2018). An output is a true positive when it is within 20 pixels of the annotation in the $256 \times 256$ images. The mean distances are calculated using only true positives. The rows are split into two settings: trained on only 64 randomly selected annotated frames, or on the full dataset. LSSL is an abbreviation for Laparoflow-SSL. All entries except the last have the same architecture. 'No SSL' was entirely randomly initialized and fine-tuned all parameters. 'LSSL side-tune' had its backbone frozen and trained a side-tune network, as in the segmentation experiments.

| #labeled *EndoVis15* | | Recall (%) / Precision (%) / Distance (px) | | | | |
| | | L Clasper | R Clasper | Head | Shaft | F1 |
| --- | --- | --- | --- | --- | --- | --- |
| 64 | No SSL | 47/65/7.8 | 50/63/7.8 | 46/71/8.4 | 43/61/8.6 | .56 |
| | Laparoflow-SSL | **98**/85/4.4 | **97**/83/5.1 | 78/86/4.2 | 64/65/5.8 | .82 |
| 940 | No SSL | 72/77/6.6 | 67/71/6.3 | 75/90/4.5 | 67/90/4.2 | .76 |
| | LSSL fine-tuned | 78/83/6.7 | 75/82/7.7 | 59/83/5.1 | 64/82/5.0 | .78 |
| | LSSL side-tune | 63/**98**/**4.2** | 62/**94**/**4.4** | 59/**98**/**3.5** | 62/**96**/**3.4** | .77 |
| | Laparoflow-SSL | **98**/92/4.4 | **97**/85/5.1 | **82**/80/4.4 | 75/71/5.8 | **.85** |
| | Du et al. (2018) | 86/87/5.0 | 85/86/5.4 | 76/77/6.6 | **91**/92/8.6 | **.85** |

Furthermore, it is remarkable that the performance degrades when all 245 images from m2caiSeg are used, even for the models without SSL pretraining. At the time of writing, we were unable to collect conclusive evidence for any explanation of this phenomenon.

When evaluating on robotic instrument pose estimation, simple networks trained on top of the representations are competitive with previous work. The advantage of pretraining is most apparent in a setting with few annotated images. Some access to the pixel values, such as through a side-tune network, seems to allow the networks to place their outputs closer to the intended ground truth.

## 4. Conclusion

We have shown that it is possible to pretrain pixel representation backbones for laparoscopic surgery in a fully self-supervised manner leveraging the availability of video. By building downstream models on top of these representation backbones, we achieve competitive results while requiring an order of magnitude less data. For semantic segmentation, this applies to both the instrument class and the tissue classes, even with only 16 annotated images for fine-tuning. This proves that self-supervised pretraining on laparoscopic video is a viable strategy to reduce the cost of collecting an annotated dataset. Our approach can perform inference in real-time and is architecture independent, which makes it complementary to future improvements on segmentation models and fine-tune protocols.

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

## Appendix A. Practical Advantages.

### A.1. At inference.

Applications built according to our proposed structure of a frozen representation backbone with simple downstream task specific heads can be both memory efficient and fast at inference. If a fast architecture was chosen for the backbone, the relatively few calculations performed by the simple head should be easy to parallelize and should not incur much additional delay. If the backbone is kept frozen during downstream task training, many downstream tasks can be performed in parallel on the same GPU, sharing most of the memory and calculations, i.e. those of the backbone.

### A.2. During development.

For the development of downstream applications, the use of our self-supervised approach allows one to start building and experimenting sooner and facilitates a quick feedback loop. Setting up a process for annotation of data can be cumbersome, especially when the required level of expertise of the annotators is high. Even after annotated data starts coming in, it can take a while before a sufficiently large body has been collected. In contrast, with our approach backbone training can start as soon as the unlabeled data is available. This could even be extended to a sort of federated learning. Analysis through clustering or dimensionality reduction of the representations can help guide the acquisition of data.

Because of the lower need for annotated data, the transition to downstream task related experimentation can be made sooner. Through the augmentations performed during the training of the backbone, the representations are robust to slight variations in the data. As is

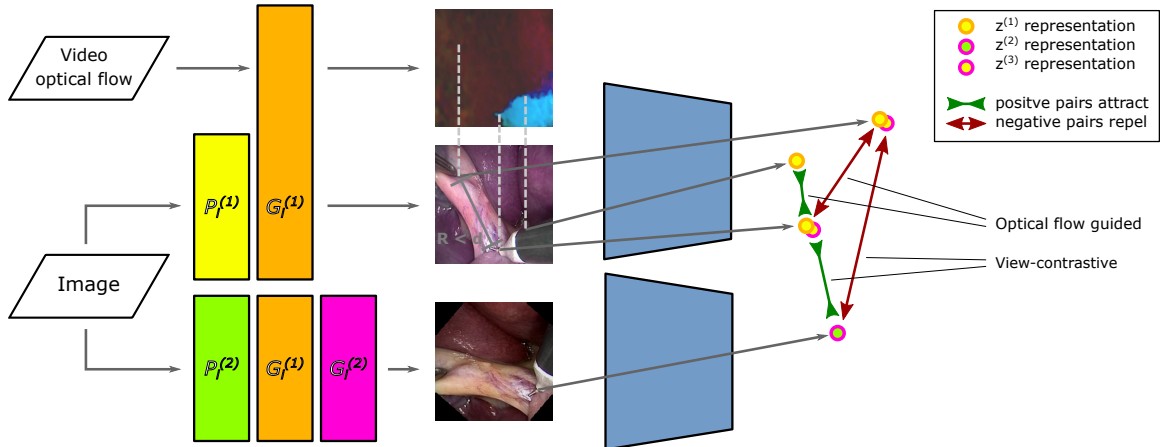

Figure 2: The steps for calculating the contrastive loss terms (Section 2.1, pages 2-4). Two views of a dataset sample are constructed by generating and applying two photo-metric ($P_i^{(1)}$ and $P_i^{(2)}$) and two geometric transformations ($G_i^{(1)}$ and $G_i^{(2)}$). The geometric transformations are also applied to the optical flow frame for easy indexing. Consecutively applying two geometric transformations, instead of one entirely independent transformation, leads to every representation $\mathbf{z}^{(2)}$ (bottom path) having a counterpart $\mathbf{z}^{(1)}$ (middle path). These counterparts are easily accessed by applying $G_i^{(2)}$ on the tensor of $\mathbf{z}^{(1)}$ representations, resulting in $\mathbf{z}^{(3)}$ representations. Note how the color of the outline of the points indicates the last applied geometric transformation. For less reliance on optical flow, the distance between pixels can also be used to choose negative pairs. See Section C.2 in the appendix.

the case in our experiments, this makes it viable to forgo augmentations during downstream training and simply precompute the representations of the annotated training data. The result is experimental runs that take up very little GPU memory and converge in a matter of minutes. Developers and researchers can quickly test several ideas in parallel and get feedback for an iterative approach. As an extension, these lightweight experimental runs can make grid search for hyperparameters or network architectures more affordable in time and hardware.

## Appendix B. Augmentations and Preprocessing.

### B.1. Optical Flow.

For pretraining, we compute optical flows and reduce redundancy by only keeping frames at 5 fps, resulting in an unlabeled dataset of 157450 images. The optical flows are computed from the videos by the NVIDIA DeepStream element Gst-nvof (NVIDIA, 2023) at the highest quality preset. To test for advantages when, in practice, one has access to a wider range of unannotated video sources, we also pretrained a backbone on the 13 M2CAI videos as

well as 13 randomly selected videos from Cholec80 that were not annotated in CholecSeg8k ('LSSL Cholec80 pretrain' entry in Table 1), resulting in a total of 327100 images in the pretraining set.

## B.2. Augmentations during pretraining.

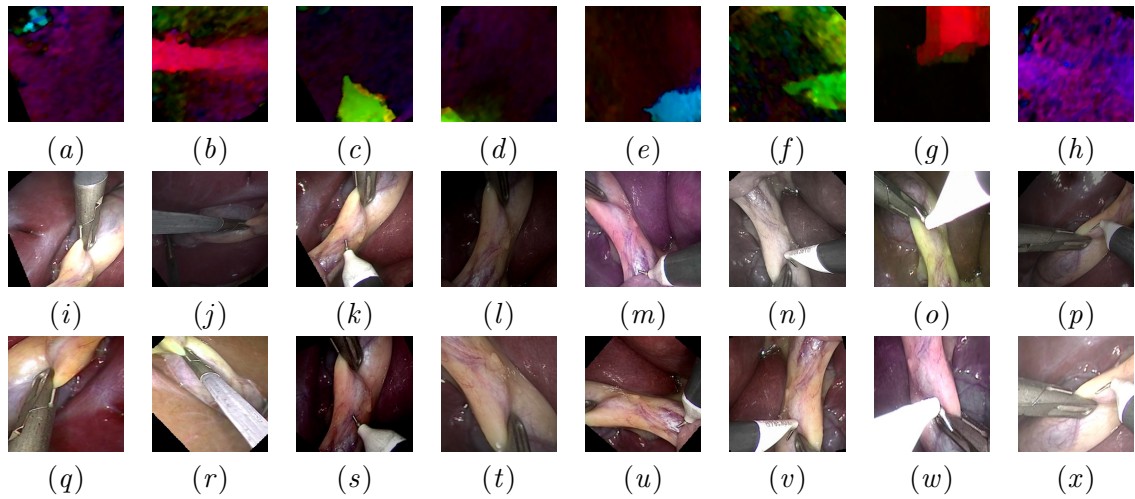

Figure 3: Some examples of views that were constructed through our augmentation pipeline. The rows correspond to the paths in Figure 2. The middle row $(i - p)$ shows the view resulting in representations $\mathbf{z}^{(1)}$. The top row $(a - h)$ shows the corresponding geometrically transformed optical flow frames. With the optical flow colors encoded in HSV, the hue represents the flow direction and the brightness the flow magnitude. The bottom row $(q - x)$ shows the view resulting in representations $\mathbf{z}^{(2)}$. Because of the composition of geometric transformations, every point in the bottom row can also be found in the middle row.

In contrastive self-supervised learning, the usefulness of the representations is heavily dependent on the transformations leading to the two contrasted views. As shown in Figure 2, when training our representation backbones, the transformations were applied in two steps. First, two different sets of photometric transformations are applied on two copies of an image from the dataset. These transformations are applied on pixel color triplets with values between $[0, 255]$ and include a random brightness shift between -64 and 64 points in RGB space, a contrast change by multiplying RGB values by a random number between 0.25 and 1.75, a saturation change by multiplying the relevant dimension in HSV space by a random number between 0.25 and 1.75, and a hue shift by adding a value between -24 and 24 degrees to the hue channel. Second, both views undergo a different geometric transformation. These transformations can include a flip, a rotation between $] - 180, 180]$ degrees, and a crop up to 150% zoom. The second crop is always taken within the first crop. This ensures a maximum amount of pixels with representations in both views, namely all

pixels in the second view. Some examples of these transformations are shown in Figure 3. The resulting images are $256 \times 256$ in size.

For easy indexing, the optical flow frame and a mask of valid pixels are geometrically transformed along with the video frames. The pixel optical flow vectors themselves do not need to be modified in the current implementation as the training is only influenced by the relative difference between vectors. In an extension, pixels of nearby frames could be contrasted. Properly transforming the flow vectors could make the sampling of such pixel pairs more efficient. Because of the endoscope, pixels are usually captured within a circular mask. Rotations can also introduce pixels with no value. To keep track of these and exclude them in the loss calculation, each video in the datasets has an associated valid pixel mask that undergoes the same geometric transformations.

## Appendix C. Variations on the loss function.

### C.1. Optical flow guided only.

To disentangle the influence of view-contrastive and optical flow guided learning, one can ablate the former in our proposed loss function for representation backbone pretraining. This can be done by using similarity function $s^{(1,1)}$ (Equations (5) and (2)) instead of $s^{(2,3)}$ in Equation (1). Naturally, it then becomes redundant to add the pixel itself to the positive set. The representation of the anchor point is trivially the same and does not need any attracting.

### C.2. View-contrastive pixel representation learning.

A view-contrastive InfoNCE loss term can be constructed to focus only on obtaining invariant representation to endoscope pose, optical settings and lighting conditions. By applying two random pairs of transformations on the same image we obtain two views of that image. The pixel pair consisting of a pixel and its counterpart in the other view is assigned a positive weight of 1. For the negative pairs, one would prefer to use pixels that belong to different semantic classes. This is unknown, however, because we do not make any assumptions about the availability of annotations. To improve the odds of using semantically different negative pairs, we set a radius in the transformed image space. All pixels within the radius of each anchor are too close. while all pixels beyond the radius are assigned a negative weight of 1 (Equations (10) and (11)). For our experiments, we set a radius of $R = 1/3$ of the image width. The pixel representations and weights are then combined into a loss term as described in Equation (1) with similarity function $s^{(2,3)}$ (Equation (5)). In Table 1, the pretraining for 'View-contrast, no flow' was done using only this loss.

$$\mathcal{P}_{i,a} = \Big\{a\Big\}, \qquad \mathcal{N}_{i,a} = \Big\{n \mid R < d_2(a,n)\Big\}, \tag{10}$$

$$w_p = w_n = 1. \tag{11}$$

### C.3. Balancing contrastive loss terms.

Although both loss terms can use the same pixel representations, the weights assigned to each triplet of pixels are very different. This affects the scale of the loss terms and the

importance of the learning signals when updating the parameters of the network with a sum of loss terms. To balance this, we estimate the difference in scale of the distributions of the weights during training. This is done by summing the total weight of all pixel triplets in the entire batch for each loss term separately. Each loss term is then multiplied by a factor equal to the total of the other loss term divided by the sum of both totals. Mathematically, this is equivalent to a weighted average of the loss terms, weighted by the reverse of their totals. Balancing allows for more interpretable control over the influence of each loss term.

## Appendix D. Downstream implementation details.

### D.1. Network architecture.

When representation backbones are frozen, side-tune networks give the downstream model access to the pixel values and the relatively shallow architecture enables the model to learn simple, downstream-specific features that might have been missed during the self-supervised pretraining. Both layers of the side-tune network for semantic segmentation are fully convolutional, consisting of 64 and 128 $3 \times 3$ kernels respectively. The 128 long pixel feature vector is concatenated to the 256 long backbone representation vector before being passed to the segmentation head. The segmentation head also consists of 2 fully convolutional layers, both with 256 $3 \times 3$ kernels. A final $1 \times 1$ convolutional layer with a softmax prepares the semantic segmentation output. We leave other fine-tuning protocols (Jia et al., 2022) to future work.

For our robotic instrument pose estimation experiments, the detection head consists of 7 fully convolutional layers, with 16 $3 \times 3$ kernels. The final layer consists of $1 \times 1$ kernels with a sigmoid activation function.

### D.2. Representation backbone pretraining.

During pretraining, the memory footprint is decreased to fit on an NVIDIA RTX 3090 by subsampling 16 anchor pixels from each image instead of using all for the loss calculation. Representation backbones are trained for 1000000 batches, showing convergence.

### D.3. Downstream training.

During training for semantic segmentation, the sum of cross-entropy and Lovász-Softmax loss (Berman et al., 2018) is calculated from the randomly selected subset of the m2caiSeg dataset. All training is done with a mini batch size of 8 and the reported numbers are at convergence. Section E in the appendix contains more comparisons and ablations.

Similarly for key point detection, as loss function, we average the binary Lovász-Softmax loss and cross-entropy for every key point type. Following Du et al. (2018) for loss targets, we constructed 2D Gaussians centered at the labelled point locations.

## Appendix E. Additional Results.

### E.1. Inference latency, throughput and memory footprint.

Profiling of our robotic instrument pose estimation implementation indicates its low memory footprint and real-time processing capabilities. The number of parameters is $3.7M$ for the

Table 3: Supplemental experiment runs to those listed in Table 1. Most notably, we list the results here for a run with a DeepLab (Chen et al., 2017) backbone architecture instead of the SegFormer (Xie et al., 2021) architecture that is used in all other instances of Laparoflow-SSL. 'No SSL, with same augmentations' starts with randomly initialized weights and also fine-tunes the weights of the backbone. During its training, the same, relatively heavy augmentations were applied as for the pretraining of the backbones (Figure 3). 'Stronger augment LSSL' has the same architecture as 'Laparoflow-SSL' but was built on top of a backbone that was pretrained with even more severe augmentations.

| #labeled m2caiSeg images | | Test on *CholecSeg8k* | | | | | | | |
| --- | --- | --- | --- | --- | --- | --- | --- | --- | --- |
| | | Dice | | | | IoU | | | |
| | | Fat | Ins | Gb | Bkg | Fat | Ins | Gb | Bkg |
| 16 | No SSL | .82 | .63 | .52 | .84 | .69 | .47 | .36 | .72 |
| | No SSL, with same augmentations | .78 | .55 | .48 | .79 | .64 | .38 | .32 | .66 |
| | Laparoflow-SSL | .84 | .71 | .61 | .88 | .72 | .54 | .44 | .78 |
| | Laparoflow-SSL, DeepLab backbone | .84 | .70 | .63 | .89 | .72 | .53 | .46 | .81 |
| 245 | Laparoflow-SSL | .84 | .64 | .58 | .86 | .73 | .47 | .40 | .76 |
| | Stronger augment LSSL | .82 | .67 | .59 | .87 | .69 | .51 | .42 | .76 |
| | Laparoflow-SSL without side-tune | .81 | .58 | .53 | .85 | .68 | .41 | .36 | .74 |
| $6k^*$ | U-Net++ (Zhou et al., 2018) | **.91** | .61 | **.63** | | | | | |
| | UNETR (Hatamizadeh et al., 2022) | .88 | **.71** | .42 | | | | | |
| | DeepLabV3+ (Chen et al., 2017) | .86 | .62 | .60 | | | | | |

Table 4: Laparoflow-SSL results on the same dataset that it was fine-tuned on.

| #labeled m2caiSeg images | | Test on *m2caiSeg* | | | | | | | |
| --- | --- | --- | --- | --- | --- | --- | --- | --- | --- |
| | | Dice | | | | IoU | | | |
| | | Fat | Ins | Gb | Bkg | Fat | Ins | Gb | Bkg |
| 16 | No SSL | .62 | .79 | .67 | .85 | .45 | .65 | **.51** | .74 |
| | Laparoflow-SSL | .56 | .82 | .64 | .81 | .39 | .70 | .47 | .69 |
| | Stronger augment LSSL | .58 | .83 | .68 | .82 | .41 | .71 | **.51** | .70 |
| 245 | Auto-encoder (Maqbool et al., 2020) | | | | | **.53** | **.73** | .50 | |

SegFormer-B0 backbone plus $20k$ for the narrow key point detection head. Implemented in pytorch 1.11.0 on top of CUDA 12.0, it took up 3160MiB on an NVIDIA RTX3090 and processed 1080p video frames one by one at $39 fps$.

## Appendix F. Visualizations.

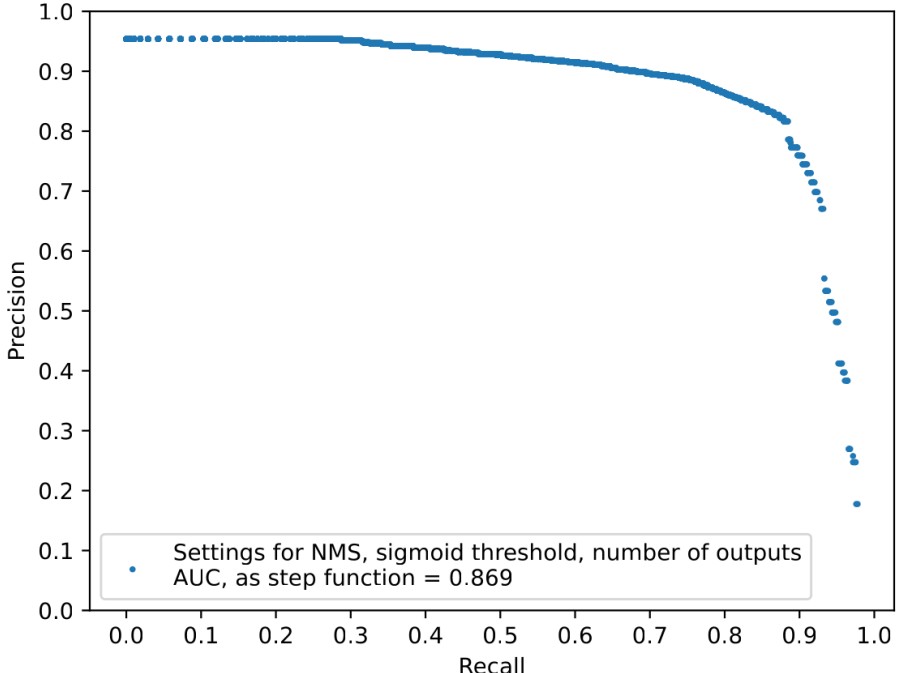

Figure 4: A precision-recall plot for the '940 Laparoflow-SSL' entry in table 2. This plot illustrates how different settings of inference hyperparameters result in a trade-off between precision and recall. The hyperparameters in question convert the sigmoid outputs of the last layer of the key point detection head into the binary prediction of the presence of key points of each type. They are the radius for the Non-Maximum Suppression, the threshold value on the sigmoid outputs for predicting a key point, and the maximum number of key points of each type that can be predicted in a single image.

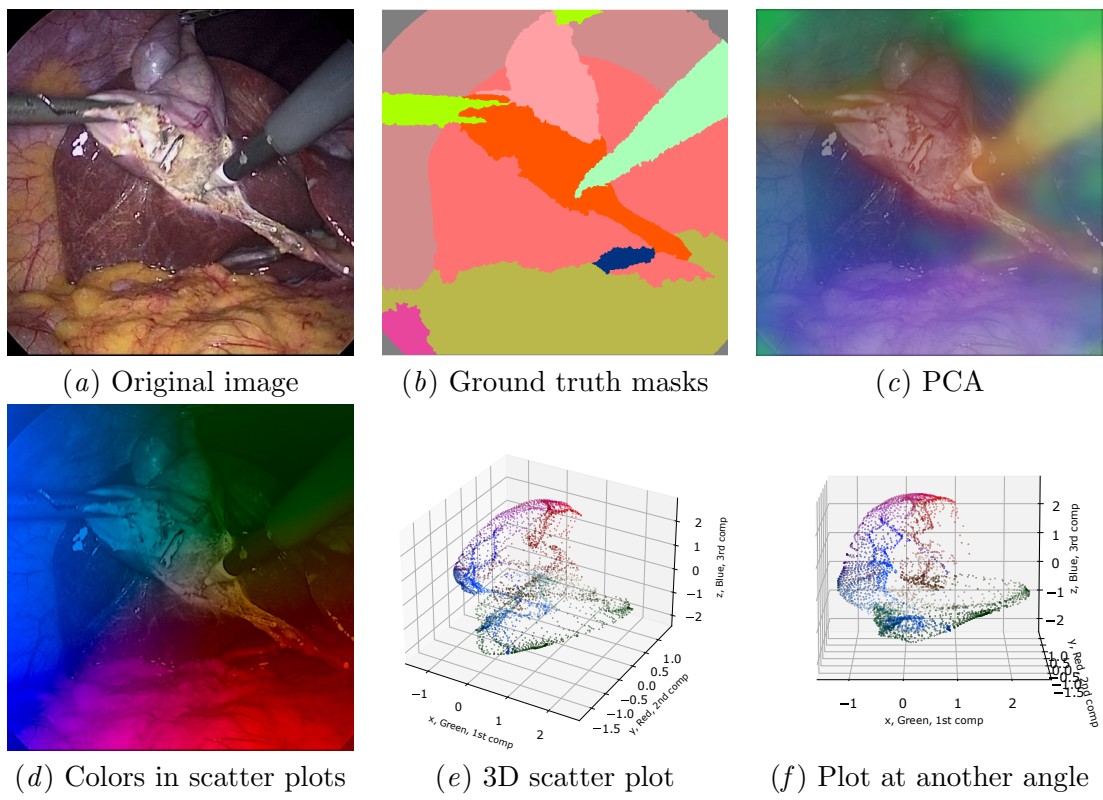

($a$) Original image     ($b$) Ground truth masks     ($c$) PCA

($d$) Colors in scatter plots     ($e$) 3D scatter plot     ($f$) Plot at another angle

Figure 5: Without access to any labels, the learned representations seem conducive to telling apart the major tissue types. ($a$) Image 15799 from video 12 in CholecSeg8k ($b$) Semantic class masks, notably: gall-bladder (light pink), liver (saturated pink), connective tissue (orange), fat (khaki green) ($c$) After pretraining, we can reduce the dimensionality using PCA to the GRB color channels. No ground truth semantic masks or explicit clustering was involved. ($d$-$f$) The bottom row shows the same dimensionality reduced representations in scatter plots. Using the color legend in ($d$), one can find the pixel representations of the fat tissue at the top. The L-hook tool on the right is roughly clustered to the right of the scatter plots.

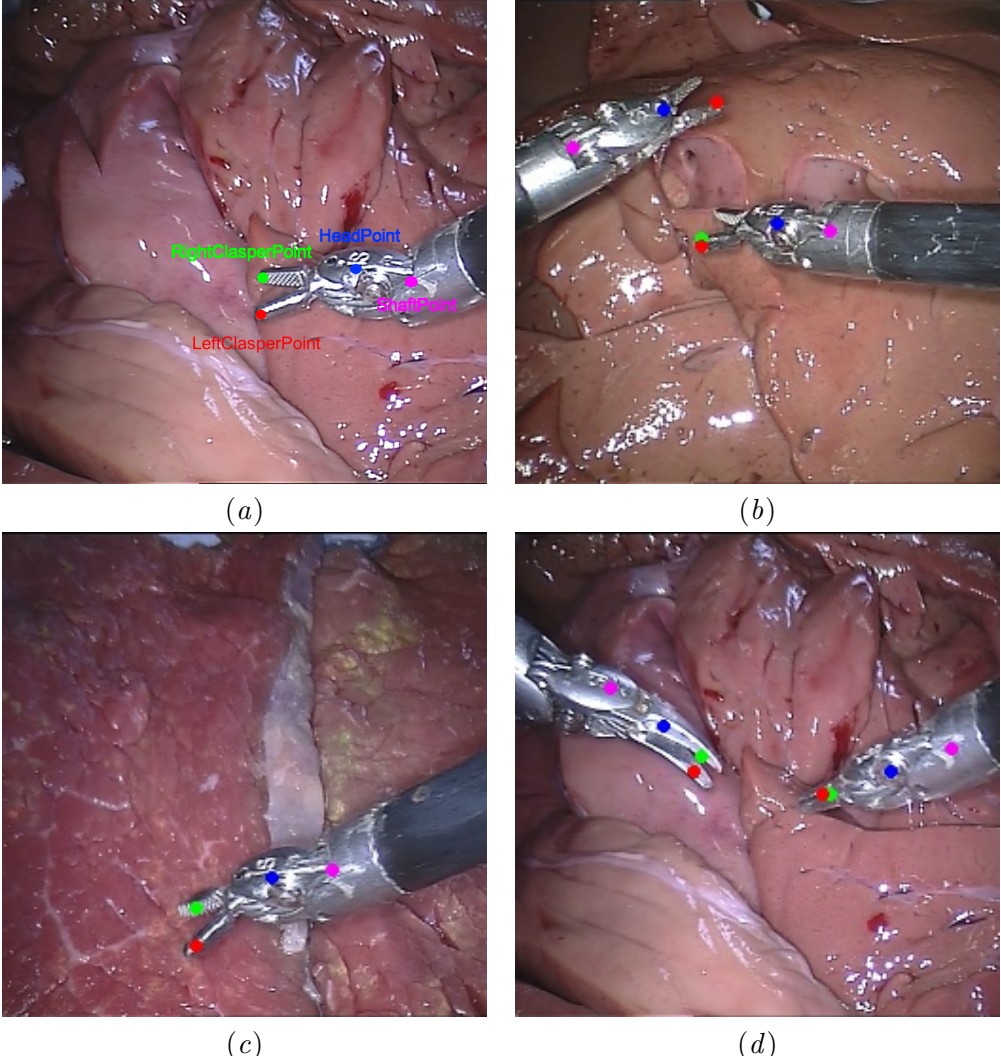

Figure 6: Some qualitative examples of robotic instrument key point detection outputs on the EndoVis15 test set (EndoVis, 2015). (*a*) The legend provided by Du et al. (2018) accompanying their dataset refinement. (*b*) Our method can handle multiple instruments in a single frame. There are indications, however, that there is confusion between Left- and RightClasperPoints for instruments on the left. Instruments mostly appear on the right in the annotated training set. Perhaps this is also related to the mirroring invariance introduced during the generic pretraining. The tool on the left in Figure (*d*) was not part of the pretraining or annotated training set.

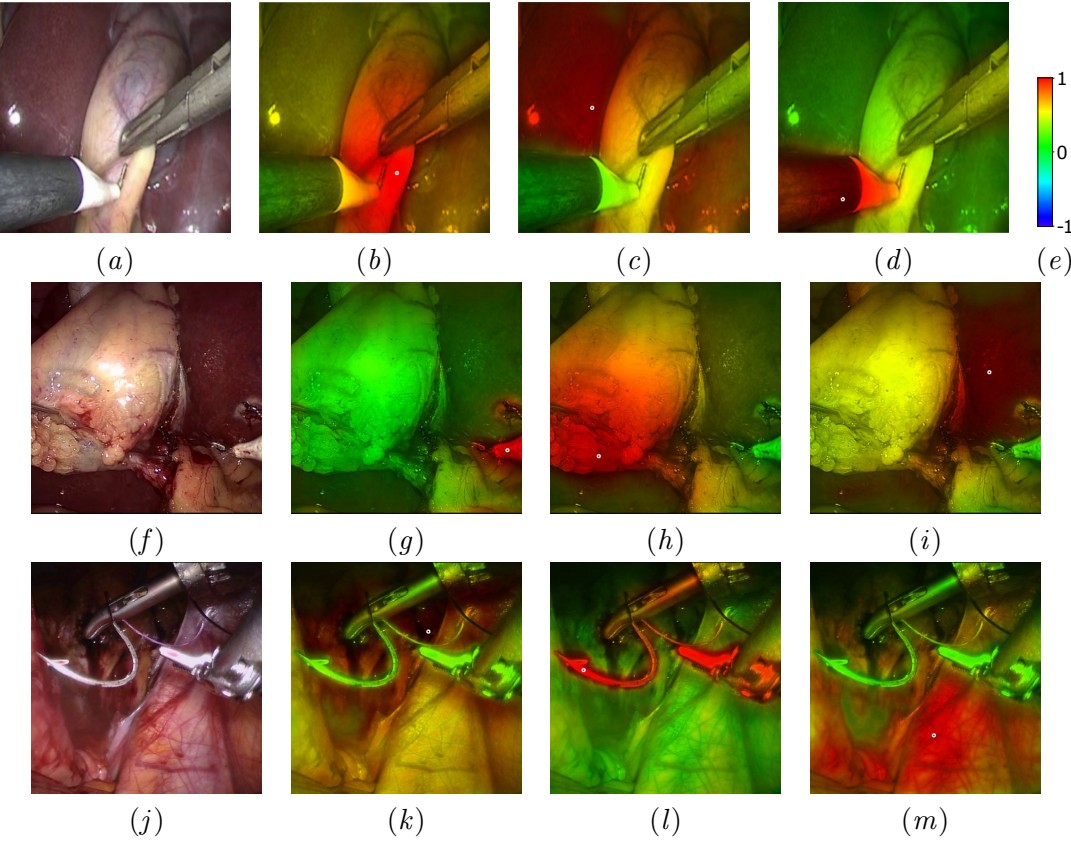

Figure 7: Cosine similarities between the representations of the anchor (white circle) and other pixels in the same image. $(a)$ The image in the top row originated from the M2CAI dataset (Twinanda et al., 2016; Stauder et al., 2016). $(f)$ The image in the middle row is from the Cholec80 dataset (Twinanda et al., 2016). $(j)$ For instances of robotic instruments, we extracted images from the RARP-45 dataset (Van Amsterdam et al., 2022). The same trained representation backbone was used for all three.

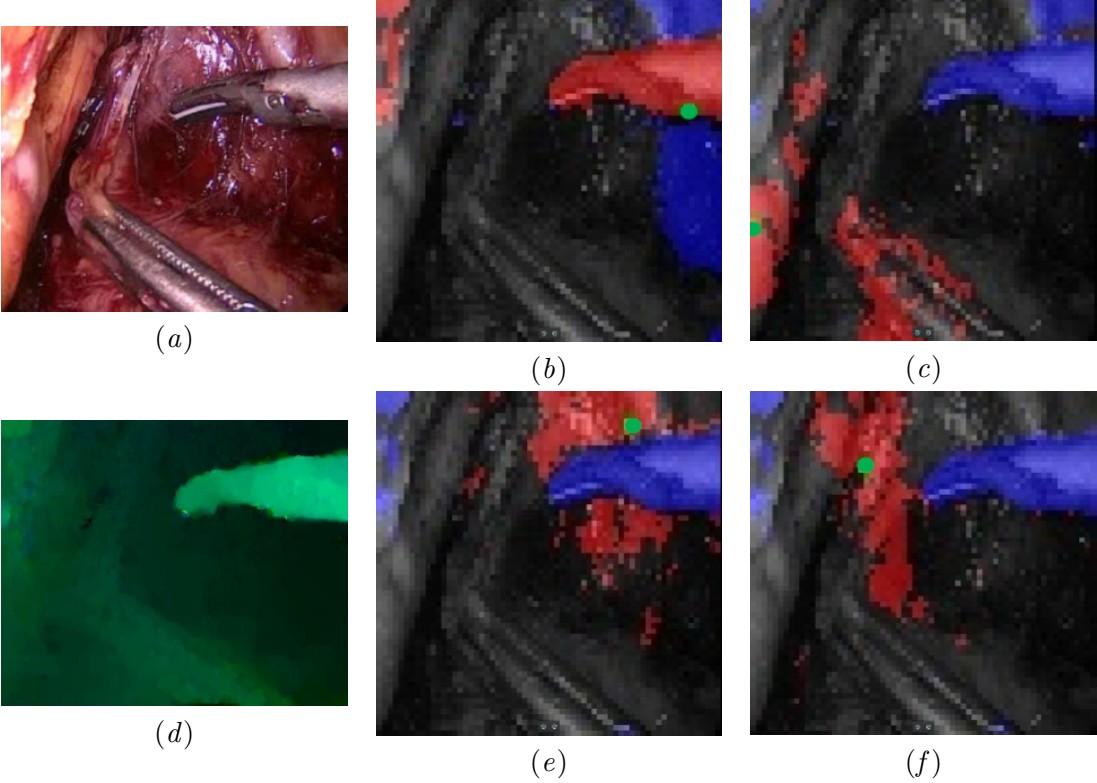

Figure 8: A visualization of the most positive (red) and negative (blue) pixels with respect to an anchor pixel (green), based on the differences in optical flow. Note that when the anchor is a tissue pixel, moving instrument pixels are almost always among the most negative. Similarly, pixels closer to the camera have larger optical flow due to camera motion and are thus more likely to have large negative weights.

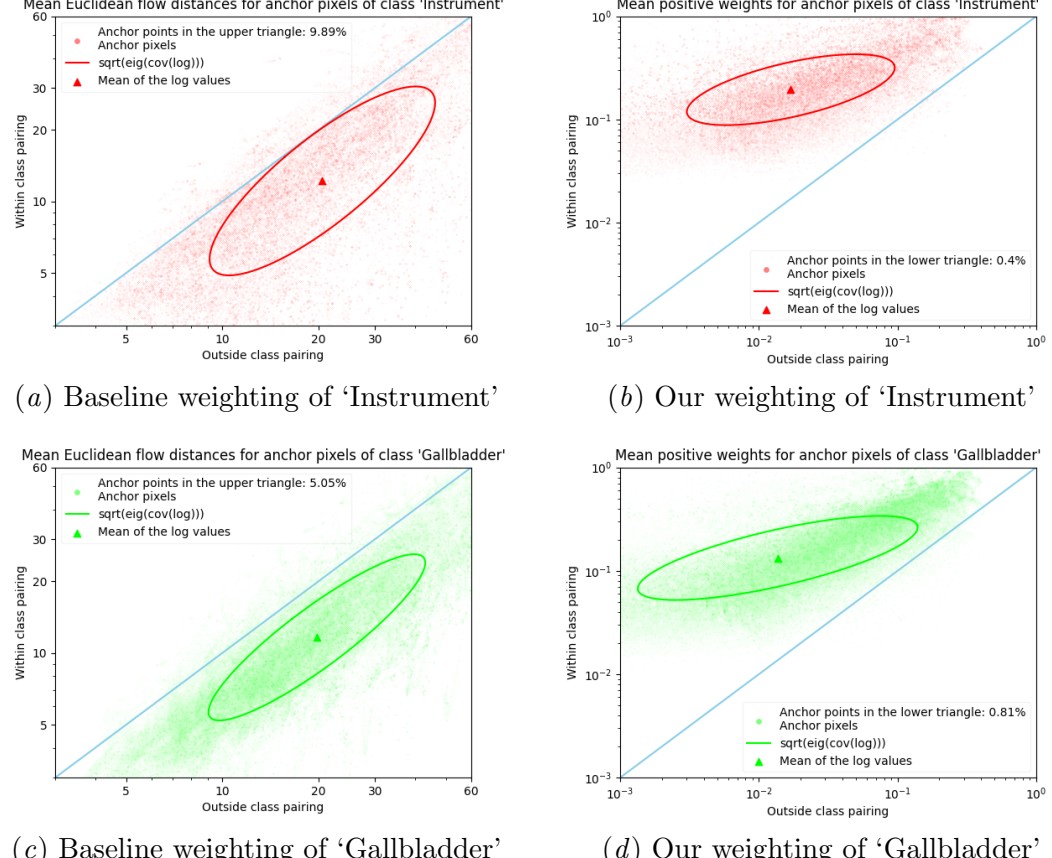

($a$) Baseline weighting of 'Instrument'      ($b$) Our weighting of 'Instrument'

($c$) Baseline weighting of 'Gallbladder'      ($d$) Our weighting of 'Gallbladder'

Figure 9: These are scatter plots of all anchor pixels in the cholecSeg8k dataset of the 'Instrument' and 'Gallbladder' classes. The position of a dot is determined by the mean values of distances or weights obtained when comparing that anchor pixel to all other pixels in its video frame, with the mean over pixels of other categories on the horizontal axis, and the mean over pixels of the same category on the vertical axis. None of the anchors in the cholecSeg8k dataset fell beneath the motionless threshold. The plots illustrate how using simple Euclidean distance between optical flow vectors would lead to a higher ratio of distractor pixels, i.e. those pixels would cause more attraction between representations of different semantic classes than between representations of the same class.

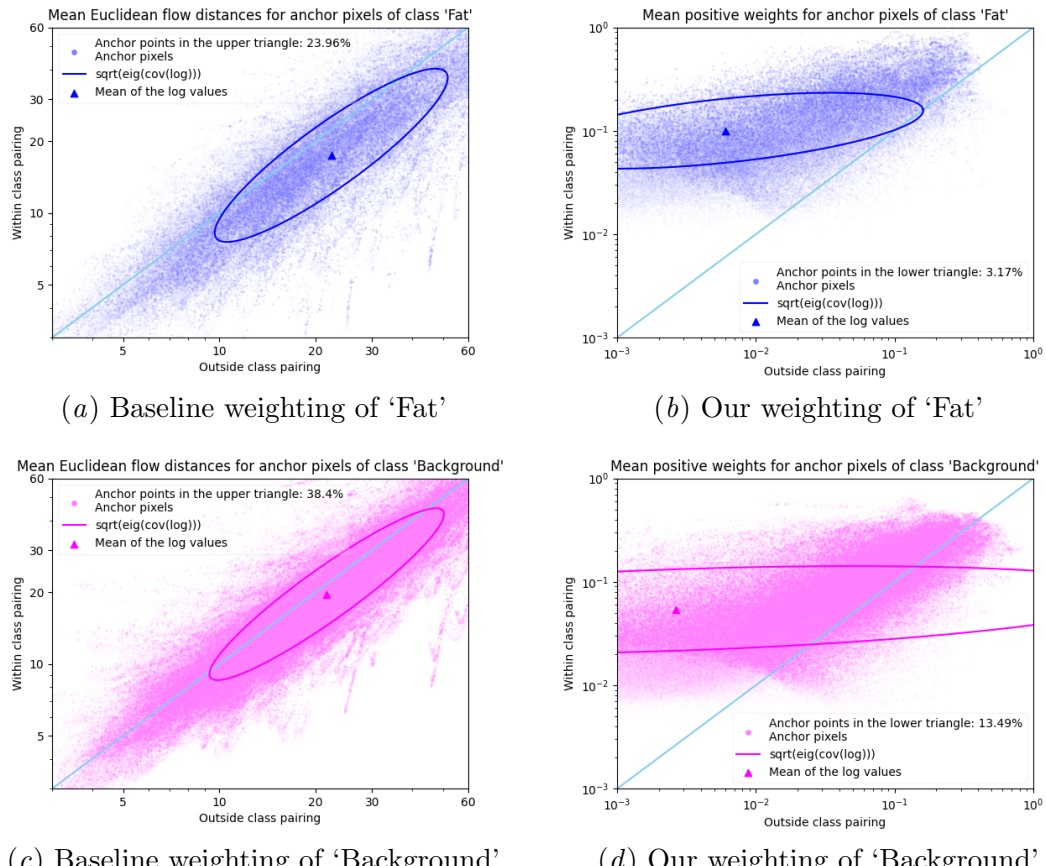

($a$) Baseline weighting of 'Fat'  ($b$) Our weighting of 'Fat'

($c$) Baseline weighting of 'Background'  ($d$) Our weighting of 'Background'

Figure 10: These are scatter plots of all anchor pixels in the cholecSeg8k dataset of the 'Fat' and 'Background' classes. The position of a dot is determined by the mean values of distances or weights obtained when comparing that anchor pixel to all other pixels in its video frame, with the mean over pixels of other categories on the horizontal axis, and the mean over pixels of the same category on the vertical axis. None of the anchors in the cholecSeg8k dataset fell beneath the motionless threshold. The same observations can be made as in the previous figure.

