# OpenReview forum: "Laparoflow-SSL: Image Analysis From a Tiny Dataset Through Self-Supervised Transformers Leveraging Unlabeled Surgical Video"
_MIDL.io/2024/Conference — MIDL 2024 Poster_

### Official Review · Reviewer_ueHF · 2024-02-27

**Confidence:** 4
**Preliminary Rating:** 5
**Recommendation:** Oral
**Final Rating:** 5

**Summary:**

The authors leveraged optical flow and Info Noise-Contrastive Estimation (InfoNCE) loss to pre-train pixel representation backbones for laparoscopic surgery in a fully self-supervised manner. They evaluate their design by solving downstream tasks on top of these representation backbones for semantic segmentation and robotic instrument key point detection.

**Strengths:**

(1) The use of optical flow during self-supervision to contrast parts of medical images and the proposed dense pixel representation contrastive loss seem to be novel.

(2) The authors evaluated two downstream tasks with their self-supervised trained backbones for semantic segmentation and robotic instrument key point detection.

(3) The authors provided many useful information in the appendix section including visualizations, more experiment results including using different backbone architectures (DeepLab instead of SegFormer), and great details and examples for data augmentation.

**Weaknesses:**

(1) The authors did not compare with other state-of-the-art self-supervised training methods in the paper.

(2) There exists some small writing issues in the paper, please refer to my detailed comments.

**Detailed Comments:**

For weakness (1),

the authors did not compare their proposed method with other state-of-the-art self-supervised methods.

For weakness (2),

(a) Please fix grammar errors throughout the paper. Here I list some examples:

-- "a factor 10"  to  "a factor of 10"

--"the pair of pixels fit as a positive pair or a negative pair" to "the pair of pixels fits as a positive pair or a negative pair"

--"SegFormer is a hybrid, lightweight Transfomer architecture"  to "SegFormer is a hybrid, lightweight Transformer architecture"

(b) Is it better to put Equation (6) after (7) and (8)?

(c) Is it better to highlight the results in all Tables? (Table 2 and Table 4 for examples)

(d) LSSL is short for Laparoflow-SSL if I understand correctly, please add this in the paper.

**Justification Of Final Rating:**

I would like to thank the authors for their revision and response. The writing issues were fixed in the revision. The authors utilize several techniques for medical images as I shared in my previous review, conduct enough experiments, and share great details in the appendix, I vote Strong accept.

**Justification Of The Preliminary Rating:**

While it would be great if the authors could compare with more state-of-the-art self-supervised methods, the authors present their ideas clearly and already conducted enough experiments in my eyes. The detailed appendix section is another highlight of the paper. Despite some minor weaknesses I mentioned in previous sections, I vote strong accept. I encourage the authors to fix writing issues in their revision.

**Questions To Address In The Rebuttal:**

It would be great if the authors could compare their method with other state-of-the-art self-supervised training methods. Please fix the small writing issues for the paper.

**Special Issue:**

Yes

---

> ### Author Response · Authors · 2024-03-18
>
> Thank you for acknowledging the novelty of our work. We intend to further clarify this novelty compared to other uses of optical flow to better convey our contributions (green text in the temporary pdf). To support the clarification, we added plots to the appendix that also illustrate the advantage over simply using the Euclidean distance between optical flows in a video frame as a proxy for semantics.
>
> We have fixed the writing issues and grammar errors. Thank you for pointing some out.
>
> Of course, we fully agree that it would be great to show a comparison with other state-of-the-art self-supervised methods. Unfortunately, time or resources are now too limited to apply these methods on our datasets and downstream tasks. We do see it as an advantage that our method is complementary to many other related self-supervised works. We propose a method for discovering positive and negative pixels pairs and this can be integrated in whatever approach that processes these pairs. Naturally, it would be interesting to study the effects of such integration.

---

> > ### Comment · Reviewer_ueHF · 2024-03-20
> >
> > One small issue to address for the final version, please delete the last blank page (page 25).

---

> > > ### Author Response · Authors · 2024-03-21
> > >
> > > I deleted the last blank page. Thank you for quickly pointing this out.

---

### Official Review · Reviewer_fcmR · 2024-03-05

**Confidence:** 2
**Preliminary Rating:** 3
**Final Rating:** 4

**Summary:**

The authors propose a new Self-Supervised Learning method for surgical video. Assuming that pixels/things that move similarly are likely to be correlated or even part of the same object, they propose a novel dense pixel representation driven contrastive loss that leverages optical flow information of different views. Specifically, given an anchor pixel in a view, it is compared to all other pixels in another view by computing the differences between the optical flow of the respective pixels. From these optical flow differences, they can extract a set of positive and negative pixels for the given anchor and finally compute the proposed dense contrastive loss.

**Strengths:**

* Even If I am not familiar with self-supervised learning for video representation learning and surgical video in general, I have found the paper well-written, pleasing to read, and quite clear.
* The main assumption of the dense contrastive loss formulation is intuitive and seems reasonable while leading to robust performances on different downstream tasks of surgical video datasets.

**Weaknesses:**

* The authors did not respect the mandatory 8 pages limit.
* Even if I am not very familiar with the literature on self-supervised video representation learning, the claim that the authors are the first to study "the use of optical flow during self-supervision to contrast parts of images" seems a bit too strong. For instance, papers [1, 2, 3] seem related to the self-supervised video representation learning based on optical flow. Therefore, I think a more complete related works would be necessary.
* It is more a discussion than a weakness, it has been shown that including more negative pairs when contrasting results in better representations. In the authors' loss formulation, it seems that they use triplets of pixel representations (1 anchor, 1 positive, 1 negative). Therefore, my question is why have authors chosen this kind of contrastive loss instead of contrastive losses that include more negative pairs such as NT-Xent loss [4]?
* It would have been interesting to see a visualization to confirm the main hypothesis that the loss formulation relies on (i.e.: "things that move similarly are likely to be correlated, perhaps even part of the same object"). For instance, for annotated frames with semantic masks, it would have been possible to measure for each anchor pixel to which extent the set of positive pixels contains pixels of the same class (purity metric). I think it would strengthen the paper and would demonstrate the soundness of this hypothesis.
* The experiments section is a bit hard to follow. I think it would make this section clearer if the authors would make a clear separation between comparisons of the final model + concurrent works and the ablation studies.

References:
* [1] Takahashi, T., Yashima, S., Ishikawa, K., Sato, I., & Yokota, R. (2023). Pixel-Level Contrastive Learning of Driving Videos With Optical Flow. In Proceedings of the IEEE/CVF Conference on Computer Vision and Pattern Recognition (pp. 3179-3186).
* [2] Sharma, Y., Zhu, Y., Russell, C., & Brox, T. (2022). Pixel-level Correspondence for Self-Supervised Learning from Video. arXiv preprint arXiv:2207.03866.
* [3] Xiong, Y., Ren, M., Zeng, W., & Urtasun, R. (2021). Self-supervised representation learning from flow equivariance. In Proceedings of the IEEE/CVF International Conference on Computer Vision (pp. 10191-10200).
* [4] Chen, T., Kornblith, S., Norouzi, M., & Hinton, G. (2020, November). A simple framework for contrastive learning of visual representations. In International conference on machine learning (pp. 1597-1607). PMLR.

**Detailed Comments:**

* The network architecture section is too detailed and could be shifted to the appendices.
* Conversely, I think Figure 2 which is the main novelty of the method would have been better in the main paper.
* Table 1 and Table 2  could be resized to avoid margin overflows.

**Justification Of Final Rating:**

The authors have adressed most of my concerns about the related works issues and performed the suggested experiment. The revised version is also clearer and better organized. I am increasing my grade to weak accept.

**Justification Of The Preliminary Rating:**

Even if I am not very familiar with the literature on self-supervised video representation learning, the claim that the authors are the first to study "the use of optical flow during self-supervision to contrast parts of images" seems a bit too strong. I expect them to give a more exhaustive literature review about self-supervised representation learning based on optical flow and at least discuss the differences with the references I have provided. Additionally, in my opinion, the experiments section is a bit messy and would require additional work to be made clearer and more concise (clear distinction between final performances and ablation studies, some parts could be moved to the supplementary like detailed network architecture, ...). Since the idea is clever but I am not sure about the novelty of this work while I think the paper could be improved, I give a Borderline. However, I am willing to change the grade if the authors address all my concerns.

**Questions To Address In The Rebuttal:**

Please see the weaknesses section

**Special Issue:**

No

---

> ### Author Response · Authors · 2024-03-18
>
> ## Visualization for the main hypothesis
> Thank you for sharing your idea for a kind of purity metric. We measured the optical flow distances and our positive weights for anchor pixels of different classes in the cholecSeg8k dataset. The visualizations were added to the appendix (green text in the temporary pdf). Indeed, they demonstrate the soundness of our main hypothesis. Through optical flow guidance, in the vast majority of cases, pixel pairs with higher positive weights turn out to be part of the same semantic class. Our transformation from Euclidean distances to positive weights further solidifies this characteristic.
>
> ## Related work and contributions
> We would also like to thank you for bringing these related works to our attention. We included them in our related work paragraph. It's true that optical flow has been used for self-supervised learning before, for example in Wang and Gupta (2015) [1]. All the mentioned works use optical flow to track pixels across video frames. This tracking then produces variations of the same point on an object through time. Forming positive pairs in this manner is relevant and complementary to our work. In fact, we consider it future work to include a term with such positive pairs in the loss function.
> The focus of this work is our method that explicitly discovers relationships between pixels within a video frame. We argue that this leads to novel and perhaps more visually distinct positive pairs.
>
> ## Discussion on negative pairs
> The choice of negative pairs is indeed one more point of further research. Another option is not to include negative pairs at all, for example by regularizing like in VICReg [2].
> For this work, we feel that a triplet-loss as a starting point is conceptually easy to grasp.
>
> Finally, we were able to make changes to fit the text within 8 pages and the tables within the margins.
>
> References:
> * [1] Xiaolong Wang and Abhinav Gupta (December 2015). Unsupervised learning of visual representations using
> videos. In Proceedings of (ICCV) International Conference on Computer Vision, pages
> 2794 – 2802.
> * [2] Bardes, A., Ponce, J., & LeCun, Y. (2022). Vicregl: Self-supervised learning of local visual features. Advances in Neural Information Processing Systems, 35, 8799-8810.

---

> > ### Comment · Reviewer_fcmR · 2024-03-21
> >
> > The authors have adressed most of my concerns about the related works issues and performed the suggested experiment. The revised version is also clearer and better organized. I am increasing my grade to weak accept.

---

### Official Review · Reviewer_4tef · 2024-03-05

**Confidence:** 4
**Preliminary Rating:** 3
**Recommendation:** Poster
**Final Rating:** 4

**Summary:**

The work studies the application of self-supervised learning (SSL) in laparoscopic videos. It employs a view-contrastive (photometrically and geometrically transformed views) self-supervised contrastive loss in which the pairs are weighted using optical flow. The SSL-trained model achieves competing performance in downstream tasks (segmentation and key point detection) when fine-tuned on very little annotated data.

**Strengths:**

1.	It addresses one of the pressing challenges (limited annotated dataset) in the surgical domain and gives insights into the advantages of employing SSL techniques.
2.	It also tried to address domain-specific challenges: It uses the relative distance between motion flows and thresholding, to cater to both significant movements (by instruments) and small movements (in tissues).
3.	The proposed method achieves completing performance on two varied downstream tasks (segmentation and key point annotation) when fine-tuned on very small (8-245 or 64 samples).

**Weaknesses:**

1.	The technical novelty seems limited. It is mostly presented as summations of prior techniques. In addition to prior works on InfoNCE loss (Oord et al., 2018, Pinheiro et al., 2020) and unsupervised learning using invariance (Cho et al., 2021), the use of cross-pixel optical flow for SSL has also been been explored [a]:

	[a] Mahendran, Aravindh, James Thewlis, and Andrea Vedaldi. "Cross pixel optical-flow similarity for self-supervised learning." Computer Vision–ACCV 2018: 14th Asian Conference on Computer Vision, Perth, Australia, December 2–6, 2018, Revised Selected Papers, Part V 14. Springer International Publishing, 2019.
2.	Lacks ablation studies on using absolute vs relative distance and w/wo thresholding to show its significance in model improvement.
3.	Lacks benchmarking against any of the other SSL techniques.

**Detailed Comments:**

1.	It is difficult to comprehend the clear/incremental technical novelty employed in this work. Re-writing “our contributions points” in the introduction or the method section to clearly state the key contributions of this work will significantly help the reader.
2.	The dataset section seems a little unclear. Better re-organization of this section will significantly improve the quality of this work. For instance, having subsections: (a) SSL pre-training), (b) Fine-tuning segmentation and (c) fine-tuning key-point detection, will give a clear picture of any data leak from the trainset to test set.
3.	In the fine-tuning stage, it is unclear how having a representation model in parallel with a side-tune model would help with model performance. Insights on how representation backbone features are combined with side-tune and how it contributes to improving the model performance could be helpful.
4.	Show qualitative analysis of downstream tasks in the main paper.
5.	Figure 1 (a) could be improved further to have a better understanding of SSL and fine-tuning.
6. While the proposed method can be adopted to train most models, details on overall implemented model size and inference time could give some insights into real-time performance.

**Justification Of Final Rating:**

This work addresses one of the pressing challenges in surgical data science (limited annotated data). While the technical novelty and dataset section was unclear in the initial draft, the revised manuscript has improved the clarity on the technical novelty and dataset usage (for training backbone, segmentation and pose estimation), showcasing the significance of this work: (i) use of optical flow information to mine positive and negative pixel pairs within video frames and (ii) achieving competitive results even when trained on a very small dataset. As such, I am convinced to improve my final rating. Limited SOTA-SSL comparison and ablation studies make it difficult for me to comprehensively benchmark this work. As such, I recommend weak acceptance.

**Justification Of The Preliminary Rating:**

In my view, this work gives key insights into using SSL in the surgical domain and addresses one of the key domain challenges (limited annotated dataset). It also shows competitive performance on two varied downstream tasks when fine-tuned with minimal data (8 - 245 samples). However, there is a lack of clarity (in writing) and justification (through ablation studies) on the technical novelty of the proposed method. Taking these into account, I recommend "Borderline".

**Questions To Address In The Rebuttal:**

1. Addressing points 1-3 in the detailed comments will significantly improve the clarity of the paper and highlight its advantages.
2. Ablation studies on using (i) absolute vs relative distance and (ii) w/wo thresholding may highlight the key incremental contributions of this work.

**Special Issue:**

No

---

> ### Author Response · Authors · 2024-03-18
>
> ## Technical novelty
> Thank you for pointing out weaknesses in our writing and clarity. We've made changes to the description of the contributions and the related work (green text in the temporary pdf). This should indeed help clarify the novelty and the different approach that needed to be taken to reach our results.
> We agree with your advice for the dataset section and have modified it accordingly.
>
> ## Effects of side-tune network \& inference performance
> Indeed, we view the study of the effects of the side-tune network and other fine-tuning protocols as an interesting follow-up. For this work, however, we feel the focus is on the self-supervised training of the representation backbone. Incidentally, an important advantage of the side-tune network is that the backbone weights remain frozen, just like with visual prompt tuning. Unlike visual prompt tuning, the representation that is fed to the task specific head can be shared among several parallel tasks. To aid readers in considerations related to downstream tasks, we added a section on model size and inference time, as you suggested.
>
> ## Comparison between absolute vs relative distances
> It's good that you brought this gap in the explanation of the utility of optical flow for contrastive triplet weighting to our attention. To cover the gap, we gathered some statistics on the cholecSeg8k dataset. We added the visualizing scatter plots to the appendix. They show that our weights based on relative distance are more in line with the annotated semantic classes. In this much smaller dataset, none of the pixels had estimated optical flow magnitudes below the threshold. The threshold was necessary, though, to avoid overflow during the processing of the videos for self-supervised pretraining.
>
> We agree, of course, that more ablations and comparisons against other SSL techniques would have been interesting. Our work discovers novel positive pairs for self-supervision. In that sense, it's compatible with many related works. For now, we leave the study of the integration in other methods to future work.

---

### Comment · Reviewer_PGJb · 2024-03-27

Summary:

This manuscript introduces a self-supervised learning (SSL) framework for laparoscopic video analysis, leveraging optical flow and InfoNCE loss to pre-train pixel representation backbones. The primary innovation claimed is the application of a dense pixel representation-driven contrastive loss for better model performance on downstream tasks such as semantic segmentation and key point detection with minimal annotated data.

Strengths:

    The approach to utilize optical flow in SSL for medical image analysis and the proposed contrastive loss methodology are conceptually novel and show promise in addressing the challenge of limited annotated datasets in the surgical domain.
    Evaluation on downstream tasks offers practical insights into the effectiveness of the SSL-trained backbones, showcasing potential real-world applicability.
    The supplementary materials provided, including extensive visualizations and experiment results, enrich the paper's contributions and offer valuable resources for further research.

Weaknesses:

    The paper's novelty is questioned due to its perceived reliance on amalgamating existing techniques rather than introducing groundbreaking methods. A clearer articulation of the unique contributions beyond the combination of established methodologies would be beneficial.
    The absence of direct comparisons with other SSL methods, especially those that inspired the current approach, hampers the ability to accurately assess the proposed method's impact and advancement over the state-of-the-art.

Detailed Comments:

    Technical Novelty and Contribution Clarification: There is a need for a more detailed discussion on how the proposed method diverges from and improves upon existing SSL approaches, specifically in leveraging optical flow for dense pixel representations. Highlighting distinct advantages or innovations could strengthen the novelty aspect.
    Comparative Analysis: Including a comprehensive comparison with other state-of-the-art SSL methods, particularly those from which the proposed method draws inspiration, is crucial. This comparison should aim to clearly demonstrate the proposed method's unique value proposition and its incremental or substantial advancements in the field.
    Writing and Organization Improvements: As previously noted, addressing minor grammatical issues and optimizing the organization of content, such as the sequencing of equations and the presentation of results in tables, would significantly enhance readability and comprehension.
    Addressing Reviewer Concerns: Incorporating detailed responses to the concerns raised regarding the paper's novelty, comparison with existing literature, and minor writing and organizational issues in the rebuttal could provide clarity and potentially sway the preliminary rating.

Preliminary Rating: Borderline

Given the conceptual promise of applying SSL with optical flow in laparoscopic video analysis, the paper holds potential. However, concerns regarding the clear articulation of novelty and the lack of comparative analysis with existing methods suggest that significant revisions are necessary to fully realize and communicate the paper's contributions and impact.

Presentation Recommendation: Poster

Recommended for MELBA Special Issue: No

---

### Meta-Review · Area_Chair_urKK · 2024-04-03

**Recommendation:** Accept (Poster)
**Confidence:** 4

**Metareview:**

The concerns raised during initial review primarily pertained to over-claiming, insufficient clarity of the writing in various parts, insufficient clarity of the contributions, as well as the lack of strong baselines and competing approaches.

While the response seems to have satisfactorily addressed the first concerns, the latter - which from my perspective is also the most concerning one, namely the lack of baselines - has not been addressed. One may argue that requesting additional experiments is undesirable (e.g., according to the TPAMI motion in other conferences), and thus, since the reviewers are satisfied this could be acceptable. On the other hand, one may arguet hat without such baselines, the manuscript simply cannot demonstrate the utility of the proposed approach compared to what is already known.

Since all reviewers now recommend acceptance (2 of whom recommend weak ratings), I am not contradiciting the reviewer verdict but will mention that the above mentioned limitation is, in fact, quite major making this manuscript a lot more borderline than the scores may suggest.

---

### Decision · Program_Chairs · 2024-04-06

Accept (Poster)